# Novel Fluorescence Probe toward Cu^2+^ Based on Fluorescein Derivatives and Its Bioimaging in Cells

**DOI:** 10.3390/bios12090732

**Published:** 2022-09-06

**Authors:** Xin Leng, Du Wang, Zhaoxiang Mi, Yuchen Zhang, Bingqin Yang, Fulin Chen

**Affiliations:** 1College of Life Sciences, Northwest University, Xi’an 710069, China; 2College of Chemistry & Materials Science, Northwest University, Xi’an 710127, China; 3Provincial Key Laboratory of Biotechnology of Shaanxi, Xi’an 710069, China; 4Key Laboratory of Resource Biology and Biotechnology in Western China Ministry of Education, Xi’an 710069, China

**Keywords:** copper ion, fluorescence probe, luminescence mechanism, detection method, bioimaging

## Abstract

Copper is an important trace element that plays a crucial role in various physiological and biochemical processes in the body. The level of copper content is significantly related to many diseases, so it is very important to establish effective and sensitive methods for copper detection in vitro and vivo. Copper-selective probes have attracted considerable interest in environmental testing and life-process research, but fewer investigations have focused on the luminescence mechanism and bioimaging for Cu^2+^ detection. In the current study, a novel fluorescein-based A5 fluorescence probe is synthesized and characterized, and the bioimaging performance of the probe is also tested. We observed that the A5 displayed extraordinary selectivity and sensitivity properties to Cu^2+^ in contrast to other cations in solution. The reaction between A5 and Cu^2+^ could accelerate the ring-opening process, resulting in a new band at 525 nm during a larger pH range. A good linearity between the fluorescence intensity and concentrations of Cu^2+^, ranging from 0.1 to 1.5 equivalent, was observed, and the limit detection of A5 to Cu^2+^ was 0.11 μM. In addition, the Job’s plot and mass spectrum showed that A5 complexed Cu^2+^ in a 1:1 manner. The apparent color change in the A5–Cu^2+^ complex under ultraviolet light at low molar concentrations revealed that A5 is a suitable probe for the detection of Cu^2+^. The biological test results show that the A5 probe has good biocompatibility and can be used for the cell imaging of Cu^2+^.

## 1. Introduction

It has long been known that copper is an important trace element and serves as a co-factor for enzymes that take part in redox and oxygen reactions [1,2]. Studies that have been conducted recently indicate that copper is closely associated with several cellular behaviors, including autophagy and apoptosis [3,4,5,6]. Copper can also induce unique cell death in characteristic pathways named cuproptosis [7,8]. Several studies demonstrated that copper is involved in the epithelial to mesenchymal transition and angiogenesis, which are considered to be related to the development and spread of cancer [9,10]. Copper is also atoxic, especially to unicellular microbes, and has been explored as an effective therapeutic agent against infectious pathogens and cancer [11,12,13,14]. Because copper can only be obtained from the external environment, its homeostasis is critical and important to maintain the normal biological activities of organisms; therefore, it is critical to develop rapid and sensitive means to determine the distribution of Cu^2+^ to protect human health and the ecological environment [15,16,17].

In recent decades, many methods for Cu^2+^ detection and measurement have been developed. However, one of the greatest drawbacks of these methods is that instruments, such as inductively coupled plasma mass spectrometry and atomic absorption spectroscopy, are always necessary when performing practical tests [18,19]. Hence, the development of more rapid, convenient, and economical methods used to analyze the copper content in different samples is significantly important for the study of health and the environment [20,21,22]. Fluorescence probes have been widely used in Cu^2+^ detection in recent years because of the advantages their rapid, sensitive, and selective detection performances [23,24,25,26,27,28]. The method has advantages of good bio-compatibility, convenient portability and easy operation in bio-sensors and bio-imaging processes [29,30,31]. Due to the development of economical probes at physiological pH values, the applications of Cu^2+^ probes in biochemical analyses has been greatly developed [32,33,34,35,36,37]. Fluorescein as a type of luminescent material with good photophysical stability has attracted considerable attention [38,39]. It can be observed that amide-modified luciferin derivatives have larger potential coordination sites, which can bind to metal ions [40]. However, the interaction mode that is helpful for us to understand the mechanism of luminescence between fluorescein and metal ions has rarely been studied or discussed [41].

In the current study, we design a new A5 fluorescence probe based on a fluorescein derivative for the measurement and detection of Cu^2+^. It is worth noting that Cu^2+^ could be visually detected selectively by using A5 in PBS buffer without the disturbance of other metal ions. Additionally, the limit detection of A5 to Cu^2+^ is much lower than the level that is reported in the literature [42]. In addition, the interaction between Cu^2+^ and the probe showed that only Cu^2+^ could open of the lactam ring. Moreover, the visual-detection experiments reveal that the A5 probe could qualitatively monitor Cu^2+^ in solution samples. More interestingly, the biological test results indicate that this probe can produce fluorescence images of Cu^2+^ in living cells.

## 2. Materials and Methods

### 2.1. Chemical Reagents

Hydrochloric acid, ethanol, sodium hydrate, hydrazine hydrate, fluorescein, copper sulfate, dimethyl sulfoxide, and 4-Bromo-2-nitrobenzaldehyde were from Aladdin Reagent Co., Ltd. (Shanghai, China). All chemical reagents were used without further purification.

### 2.2. Apparatus and Instrumentation

A HITACHI F-4500 fluorescence spectrophotometer was obtained from Hitachi, Ltd. (Tokyo, Japan). Bruker Tensor 27 spectrometer was obtained from Bruker Corporation (Karlsruhe, Germany). Bruker micro TOF-Q II ESI-TOF LC/MS/MS spectroscopy was obtained from Bruker Corporation (Karlsruhe, Germany). Varian INOVA-400 MHz spectrometer (400 MHz) was obtained from Varian, Inc. (Palo Alto, CA, USA). Spectra max190-Molecular Devices was obtained from Molecular Devices Corporation (Sunnyvale, CA, USA) and Olympus FV1000 confocal microscopy was obtained from Olympus Corporation (Tokyo, Japan).

### 2.3. The Synthesis of A5

According to the literature [42], we synthesized fluorescein hydrazine from fluorescein and hydrazine. Dissolve fluorescein hydrazine (3.39 g, 9.78 mmol) and 4-bromodinitrobenzaldehyde (1.50 g, 6.52 mmol) in 50 mL of ethanol, reflux for 3 h, and then cool to room temperature. After filtering, we rinsed it with alcohol several times to obtain a yellow solid, which was stored at 5 °C for further use. ^1^H NMR (400 MHz, DMSO-d_6_) δ 9.96 (s, 2H), 9.13 (s, 1H), 8.17 (d, J = 2.0 Hz, 1H), 8.00–7.90 (m, 2H), 7.69–7.56 (m, 3H), 7.12 (d, J = 7.5 Hz, 1H), 6.66 (d, J = 2.3 Hz, 2H), 6.48 (t, J = 8.7, 5.5 Hz, 4H). ^13^C NMR (100 MHz, TMS, DMSO-d_6_) δ 164.7, 159.2, 152.5, 151.4, 148.9, 141.6, 137.0, 135.0, 129.7, 129.0, 128.4, 128.3, 127.7, 124.3, 123.9, 123.4, 113.0, 109.9, 103.1, 65.8, 56.5, 40.6, 40.4, 40.2, 40.0, 39.8, 39.6, 39.4, 19.0. MS (ESI) *m*/*z* A5 calcd. for C_27_H_16_BrN_3_O_6_ (M + Na)^+^: 580.0115, found 580.0103.

### 2.4. Colorimetric Determination of Copper Ions

A total of 1 mM stock solutions was prepared with an A5 probe, EtOH, and deionized water. During the titration tests, Cu^2+^ and 1.0 mL of 200 μM of the probe were mixed and then filled to 10 mL in a volumetric tube with PBS. In the interference test, 20 μM of Cu^2+^ and 1.0 mL of A5 (200 μM) were mixed with 1.0 mL of test substance (400 µM), and PBS was charged into a 10 mL volume tube. During the ethylenediamine titration assay, 1.0 mL of a 200 µM A5 probe, 1.0 mL of Cu^2+^ (400 µM), and various quantities of ethylenediamine were filled up to 10 mL with PBS in a volumetric tube. A total of 1 mL aliquots were injected into a 1 cm cuvette for spectroscopic analysis. A 5 nm band-pass filter was used as excitation and emission wavelengths. Absorbance was recorded at 440 nm and fluorescence intensity was recorded at 525 nm in various assays, respectively.

### 2.5. Detection Limit of Probe

From the measurement of the fluorescence signal, the detection limit was determined. To measure the δ/S ratio, the luminescence intensity of A5 (20.0 µM) was performed 10 times, and the standard deviation of the blank assay was determined. In this case, in the range of 10.0–40.0 µM, the relative luminescence intensity (525 nm) presented a good linear relationship with the concentration of Cu^2+^. The detection limit was recorded according to the following formula: detection limit = K × δ/S, δ was the standard deviation of the blank determination; S was the gradient of the concentration and intensity of the sample. Fluorescence analysis showed Y= 134.36X − 267.38 (R^2^ = 0.9941), δ = 4.926 (N = 10), S = 134.36, K = 3; and LOD = 3 × 4.926/134.36 = 0.11 µM.

### 2.6. Cytotoxicity Study

Cytotoxicity experiments were performed using the CCK-8 method. The cells were placed in a 96-well plate and incubated at 37 °C for 24 h before adding the probe at different concentrations (0 μM, 2.5 μM, 5 μM, 10 μM, 20 μM, 40 μM) and then incubated within 24 h. CCK-8 was added to each well, followed by an additional 2-h incubation. Absorbance at 450 nm was determined. All experiments were repeated 3 times and expressed as the percentage of control cells.

### 2.7. Cell Culture Experiments

MCF-7 cells were obtained from the Laboratory Center of Shaanxi Province People’s Hospital. MCF-7 cells were grown on glass-bottom culture dishes using Dulbecco’s Modified Eagle Medium (DMEM) supplemented with 10% (*V*/*V*) fetal bovine serum (FBS) and 50 μg/mL penicillin-streptomycin at 37 °C in a humidified atmosphere with 5% CO_2_ and 95% air. The growth medium was then removed and washed three times with FBS. The cells were pretreated with a 10.0 μM A5 probe for 30 min at 37 °C, washed with PBS (pH 7.4) twice and imaged. Then, the cells were incubated with 30.0 μM CuCl_2_ for 30 min at 37 °C, washed with PBS (pH 7.4) twice and imaged. Finally, the cells mentioned above were supplemented with 30 μM ATP for another 30 min and imaged.

## 3. Results and Discussion

### 3.1. Effect of the pH and Response Time

At present, it is known that under different pH conditions the spiro structure can switch between open and closed rings. Accordingly, the influence of pH on A5 to Cu^2+^ was studied in the PBS buffer (10 mM, pH = 7.4)/ EtOH (1:1, *v*/*v*) (Figure 1). It was clear that the selectivity of A5 for Cu^2+^ was slightly affected, shifting from pH 6.0 to 9.0 (Figure 1A). Therefore, it was suitable for A5 to perform the bioimaging experiment at pH 7.4. Following the addition of Cu^2+^ (20.0 μM), the fluorescence intensity at 525 nm was strengthened and reached a plateau after 120 s, which meant that Cu^2+^ could be rapidly detected by the A5 probe (Figure 1B).

### 3.2. Probe Selection and Competition

In order to evaluate the selectivity and anti-interference ability of the A5 probe in relation to Cu^2+^, selective and competitive experiments were performed on the A5 probe in PBS buffer (10 mM, pH7.4)/EtOH (1:1, *v*/*v*). As presented in Figure 2, there are no obvious absorption peaks, and emission peaks can be observed in the A5 solutions before Cu^2+^ is added. However, a distinct absorption peak and strong fluorescence were observed when Cu^2+^ was added to the solution (Figure 2A,B). Additionally, we validated the fluorescence properties via the competition experiment. After adding other ions to the solution, the fluorescence intensity slightly changed (Figure 2C). Therefore, A5 can be used as a selective probe for Cu^2+^.

### 3.3. Qualitative and Quantitative Studies 

Different molalities of Cu^2+^ (0–100 μM) were added to the solution of the A5 probe (20 μM). It can be observed in Figure 3 that the absorption of the solution gradually increases at 440 nm; meanwhile, the fluorescence intensity at 525 nm significantly increases when the Cu^2+^ concentration increases, and the fluorescence intensity increases to the highest value when the Cu^2+^ concentration reaches 3.5 equivalent. There was a good linear relationship between the fluorescence intensity and concentration of Cu^2+^ in the range of 0.1–2.0 equivalent (Appendix A). The LOD of the A5 probe for Cu^2+^ was calculated to be 0.11 µM. The experiment results show that the A5 probe has good sensitivity for the determination of Cu^2+^ content in the relevant samples.

### 3.4. Proposed Sensing Mechanism 

In order to better understand the interaction between the A5 probe and Cu^2+^, certain methods, such as Job’s plots, MS analysis, and FT-IR, were used to study it. In the ethylenediamine titration, the reaction of A5–Cu^2+^ showed that the complex reaction was reversible (Figure 4A). The Job’s plots presented a 1:1 ratio between A5 and Cu^2+^ (Figure 4B).

In addition, in the mass spectrum, a new coordination signal was observed between the A5 probe and Cu^2+^ at the position of *m*/*z* 656.9253 [C_27_H_19_BrClCuN_3_O_6_ (M + CuCl)]^+^, which further indicated a 1:1 coordination (Figure 4C). Infrared spectral analysis showed that the peaks at 1704 cm^−1^ of the probe disappeared after the complexation reaction between A5 and Cu^2+^, which indicates that the formation of Cu-O (Figure 4D).

Above all, a plausible reaction mechanism for the complexation reaction between A5 and Cu^2+^ is shown in the paper (Figure 4E).

### 3.5. The Visual Detection of the Test Strips

In order to promote good field detection, we developed test strips with the A5 probe (200.0 μM). Subsequently, it was immersed in different metal-ion solutions (200.0 μM) (K^+^, Na^+^, Li^+^, Ca^2+^, Ag^+^, Mg^2+^, Cd^2+^, Mn^2+^, Ni^2+^, Cu^2+^, Ba^2+^, Zn^2+^, Pb^2+^, Pd^2+^, Hg^2+^, Sn^4+^, Cr^3+^, Fe^3+^, Fe^2+^, Al^3+^). Interestingly, only aqueous solutions of Cu^2+^ produced a color change, especially under UV light that was visible to the naked eye (Figure 5). As shown in Figure 5, Appendix A, the test strips mixed with only Cu^2+^ presented an obvious change from ambient (Figure 5A, Appendix A) to UV light (Figure 5B, Appendix A).

### 3.6. Cell Imaging

Based on the excellent characteristics of A5, we studied the property of A5 bio-imaging in cells. First, MCF-7 cells were tested in vitro by the MTT method. MCF-7 cells were cultured with various concentrations of A5 (0–40 μM) for 24 h, showing that the A5 probe had lower cytotoxicity levels (Figure 6A). To further examine the bioimaging performance of the A5 probe, the cells with A5 and MCF-7 were incubated for 30 min, and no obvious fluorescence was observed. Subsequently, the cells were treated with Cu^2+^ (40 μM) for 1 h at 37 °C, and the results show that the inner region of the cells have obvious fluorescence. At the same time, clear bright-field and fluorescent imaging of the cells was performed, further demonstrating the good bio-compatibility of the A5 probe and its tracking effect on Cu^2+^ in cells (Figure 6B).

## 4. Conclusions

This study introduced a new “Off-On” A5 fluorescence probe, which was more selective and sensitive for Cu^2+^ detection than for other ions. At the same time, the proposed response mechanisms of A5 and Cu^2+^ were analyzed by methods, such as Job’s plot, mass experiment and infrared spectroscopy. The results show that Cu^2+^ was partially coordinated with Schiff base and the fluorescein amide carbonyl group could induce fluorescence emissions, which was helpful for us to understand the combination mode and to design a new effective probe for Cu^2+^ detection. The test-strips experiment showed that A5 can qualitatively detect Cu^2+^ in an aqueous solution. In addition, the result of the cell imaging reveals that the A5 probe has good bio-compatibility and can be used as a sensor material for Cu^2+^ in biological samples.

## Figures and Tables

**Figure 1 biosensors-12-00732-f001:**
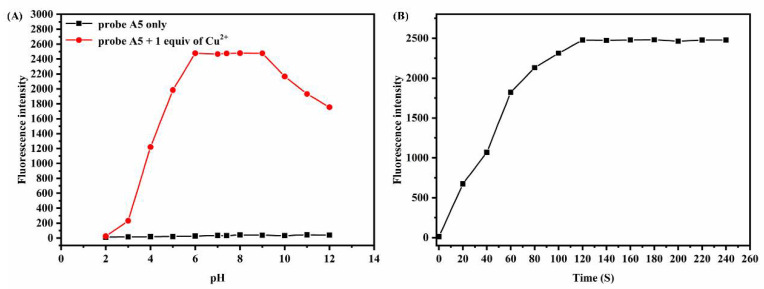
(**A**) Fluorescence intensity (525 nm) of A5 probe (20.0 µM) and the mixture at different pH levels. (**B**) Fluorescence intensity (525 nm) after adding Cu^2+^ (20.0 μM) to A5 (20.0 µM) at different times in PBS buffer (10 mM, pH = 7.4)/EtOH (1:1, *v*/*v*), λex = 440 nm, λ_ex_ = 440 nm.

**Figure 2 biosensors-12-00732-f002:**
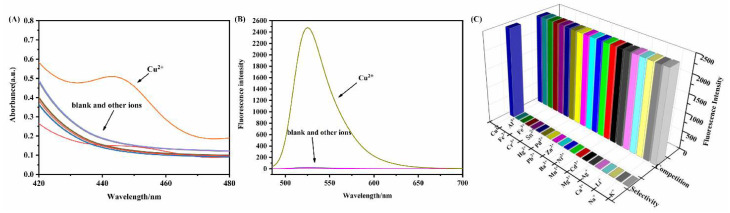
(**A**) Absorption spectrum of A5 (20.0 µM) in the presence of various metal ions: K^+^, Na^+^, Li^+^, Ca^2+^, Ag^+^, Mg^2+^, Cd^2+^, Mn^2+^, Ni^2+^, Ba^2+^, Zn^2+^, Pb^2+^, Pd^2+^, Hg^2+^, Sn^4+^, Cr^3+^, Fe^3+^, Fe^2+^, Al^3+^, and Cu^2+^ (20.0 µM) in PBS buffer (10 mM, pH = 7.4)/EtOH (1:1, *v*/*v*), λ_ex_ = 440 nm. (**B**) Fluorescence spectrum of A5 (20 µM) in the presence of various metal ions: K^+^, Na^+^, Li^+^, Ca^2+^, Ag^+^, Mg^2+^, Cd^2+^, Mn^2+^, Ni^2+^, Ba^2+^, Zn^2+^, Pb^2+^, Pd^2+^, Hg^2+^, Sn^4+^, Cr^3+^, Fe^3+^, Fe^2+^, Al^3+^, and Cu^2+^ (20.0 µM) in PBS buffer (10 mM, pH = 7.4)/EtOH (1:1, *v*/*v*), λ_ex_ = 440 nm. (**C**) Fluorescence spectrum of A5 (20.0 µM) and Cu^2+^ (20.0 µM) in the absence and presence of various metal ions: K^+^, Na^+^, Li^+^, Ca^2+^, Ag^+^, Mg^2+^, Cd^2+^, Mn^2+^, Ni^2+^, Ba^2+^, Zn^2+^, Pb^2+^, Pd^2+^, Hg^2+^, Sn^4+^, Cr^3+^, Fe^3+^, Fe^2+^, Al^3+^, and Cu^2+^ (40.0 µM) in PBS buffer (10 mM, pH = 7.4)/EtOH (1:1, *v*/*v*), λ_ex_ = 440 nm.

**Figure 3 biosensors-12-00732-f003:**
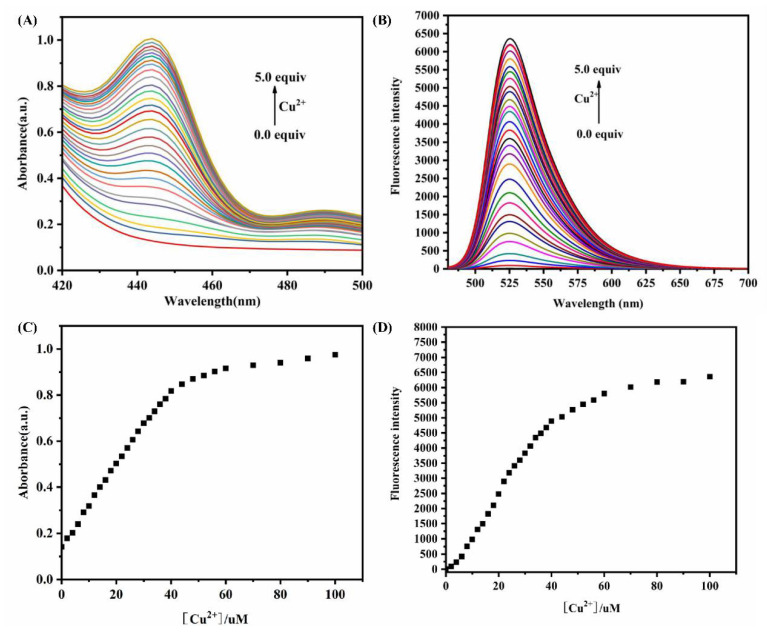
(**A**,**B**) Absorption and fluorescence spectra of A5 (20.0 µM) in the presence of different concentrations of Cu^2+^ (0.0 to 100.0 μM), λ_ex_ = 440 nm. (**C**,**D**) Plots of absorption and fluorescence intensities at 440 nm and 525 nm, respectively, with Cu^2+^ concentrations in range of 0.0–5.0 equiv. All measurements were obtained using PBS buffer (10 mM, pH = 7.4)/EtOH (1:1, *v*/*v*).

**Figure 4 biosensors-12-00732-f004:**
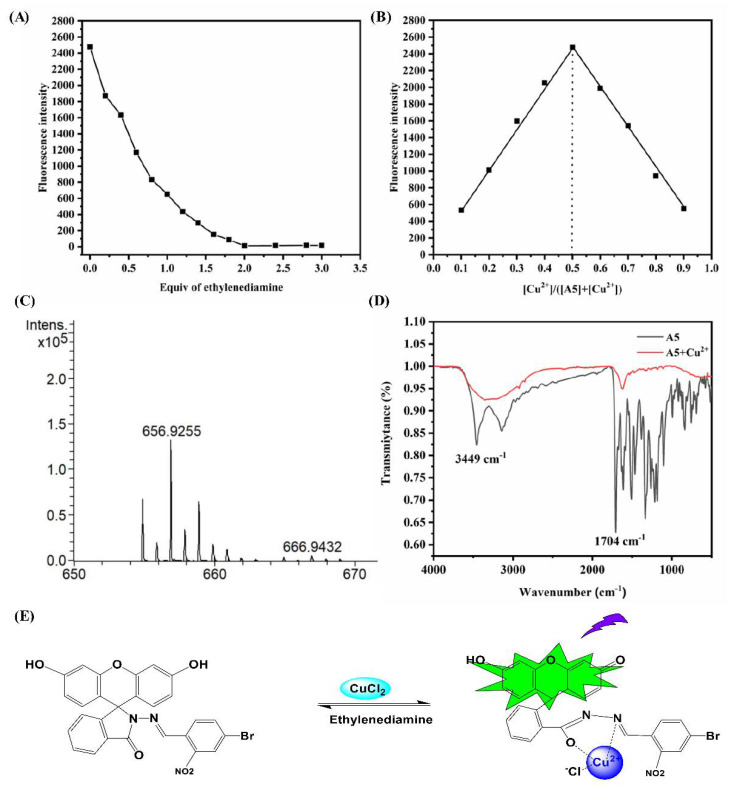
(**A**) The titration experiment of ethylenediamine and A5-Cu^2+^ in PBS buffer (10 mM, pH = 7.4)/EtOH (1:1, *v*/*v*), λ_ex_ = 440 nm. (**B**) Job’s plot of A5 probe and Cu^2+^ in PBS buffer (10 mM, pH = 7.4)/EtOH (1:1, *v*/*v*), λ_ex_ = 440 nm. (**C**) The MS analysis of A5-Cu^2+^ complex. (**D**) FT-IR spectra of A5-Cu^2+^ complex. (**E**) The proposed response mechanism of A5 with Cu^2+^.

**Figure 5 biosensors-12-00732-f005:**
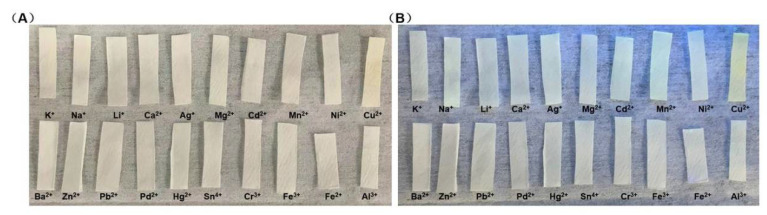
(**A**) Photographs of test strips immersed in aqueous solutions of different analytes in ambient light. (**B**) Photographs of test strips immersed in aqueous solutions of different analytes under a 365 nm UV lamp.

**Figure 6 biosensors-12-00732-f006:**
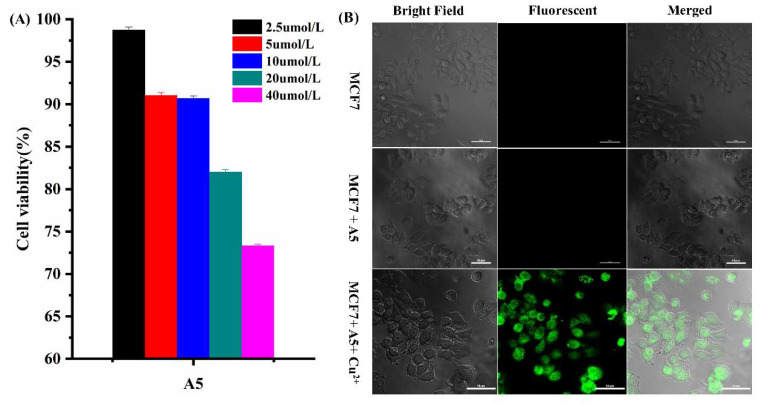
(**A**) MTT assay of MCF-7 cells in the presence of different concentrations of A5 (2.5 µM; 5 µM; 10 µM; 20 µM; 40 µM). (**B**) Bioimaging of MCF-7 cells following incubation with A5 (40.0 µM) in the absence and presence of Cu^2+^ (40.0 µM).

## Data Availability

Not applicable.

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
