# Peer review of "Novel Fluorescence Probe toward Cu2+ Based on Fluorescein Derivatives and Its Bioimaging in Cells"

_biosensors, 2022, doi:10.3390/bios12090732_

Round 1

Reviewer 1 Report

Leng et al. report a novel Cu2+ fluorescent probe A5 based on fluorescein derivatives. As the authors indicated in the introduction part, copper is an important metal ion and is involved in many biological functions. Although several fluorescent probes for Cu2+ were developed, more selective and sensitive probes are required. Fluorescent probe A5 is based on a fluorescein structure, it showed selective and sensitive fluorescence enhancement with Cu2+. Test strips experiments and cellular imaging showed the utility of A5 for a Cu2+ fluorescent probe. However, several points should be explained before publication in biosensors.

Specific criticisms:

1) UV-vis spectral data of A5 should be added in the manuscript.

2) About Figure 1A, did you conduct all of this using PBS? The method of the experiment should be accurately described.

3) The reviewer think the fluorescence and UV-vis data of A5 with monovalent copper Cu(Ⅰ) is essential.

4) About Figure 2(B), the spectral changes of each metal should be added. It is difficult to understand if other metals really interfere with Cu2+.

5) The method of cell experimentation should be included in the text. Or, it should clearly state what is stated in the SI. Why did the authors choose MCF7 cells? Cytotoxicity of A5 in MCF7 was observed at 20 μM, s there any problem with the experiment?

Author Response

Dear Reviewers:

Thank you for your comments concerning our manuscript entitled “Novel Fluorescence Probe toward Cu2+ Based on Fluorescein Derivatives and Its Bioimaging in Cell” (ID: biosensors-1853251). Those comments are all valuable and very helpful for revising and improving our paper, as well as the important guiding significance to our researches. We have studied comments carefully and have made correction which we hope meet with approval. Revised portion are marked up using the “Track Changes” in the paper. The main corrections in the paper and the responds to the comments are as follow:

  • Response to comment: UV-vis spectral data of A5 should be added in the manuscript.

Response: It is really true as reviewer suggested that UV-vis spectral data of A5 should be added in the manuscript. We have added UV-vis spectral of A5 in the revised manuscript according to the suggestion.

  • Response to comment: About Figure 1A, did you conduct all of this using PBS? The method of the experiment should be accurately described. 

Response: We are very sorry for our negligence of describe the method of experiment about the figures. About Figure 1A, we didn’t conduct all of this using PBS. And we supplement those descriptions of figures in the revised manuscript.

  • Response to comment: The reviewer think the fluorescence and UV-vis data of A5 with monovalent copper Cu(â… ) is essential.

Response: Considering the Reviewer’s suggestion, we have provided fluorescence and UV-vis spectrum of A5 in Figure (2), Figure (3) and Figure (4A, 4B) as the evidence for the interaction of A5 with Cu2+ in the revised manuscript.

  • Response to comment: About Figure 2(B), the spectral changes of each metal should be added. It is difficult to understand if other metals really interfere with Cu2+.

Response: We are very sorry for our unclearly describe about the Figure2(B). As the description of selectivity and competition histogram shows that A5 (20.0 µM) and Cu2+ (20.0 µM) in the absence and presence of various metal ions (40.0 µM) in PBS buffer (10 mM, pH = 7.4)/ EtOH (1:1, v/v), λex = 440 nm.

  • Response to comment: The method of cell experimentation should be included in the text. Or, it should clearly state what is stated in the SI. Why did the authors choose MCF7 cells? Cytotoxicity of A5 in MCF7 was observed at 20 μM, s there any problem with the experiment?

Response: We have made correction according to the Reviewer’s comments, and added the method of cell experimentation in the text as “2. 7 Cell Culture Experiments” in the revised manuscript. As we want to apply the probe in cancer study, so choose MCF7 cells which is more close to cancer cells’ sample. We are very sorry for our incorrect writing the concentration of probe A5 which should be 40.0 μM in our experiment record.

Thank you very much for your comments and suggestions.

We tried our best to improve the manuscript and made some changes in the manuscript. These changes will not influence the content and framework of the paper. And here we did not list the changes but marked up using the “Track Changes” in revised paper.

We appreciate for Reviewers’ warm work earnestly, and hope that the correction will meet with approval.

Once again, thank you very much for your comments and suggestions.

Yours sincerely,

Xin Leng

Reviewer 2 Report

Xin et al prepared a manuscript entitled “Novel Fluorescence probe toward Cu2+ based on fluorescein derivatives and its bioimaging in cell”. In this study, the authors tried to synthesis a fluorescein-based probe for copper detection and tested their bioimaging performance. After careful evaluation of the whole manuscript, I decided not to recommend this work for publication. The following are the major concerns.

1.     In this work, it is claimed that a novel fluorescein probe was prepared but the synthesis was carried out based on a published report.

2.     There are many probes available in the literature for copper sensing and this study lacks the novelty.

3.     The probe achieved high LOD compared with the published reports.

4.     Providing the evidence for the formation of Cu-O is important but none of the spectral data highlight this data.

5.     Clear explanation of the results and discussion are not presented in the manuscript. The provided spectral and data figures are of low resolutions and are not clearly visible.

6.     The imaging applications utilized high concentrations of the probe.

7.     The manuscript has grammatical and formatting errors. Overall, this manuscript is not suitable for the publication.

Author Response

Dear Reviewers:

Thank you for your comments concerning our manuscript entitled “Novel Fluorescence Probe toward Cu2+ Based on Fluorescein Derivatives and Its Bioimaging in Cell” (ID: biosensors-1853251). Those comments are all valuable and very helpful for revising and improving our paper, as well as the important guiding significance to our researches. We have studied comments carefully and have made correction which we hope meet with approval. Revised portion are marked up using the “Track Changes” in the paper. The main corrections in the paper and the responds to comments are as follow:

  • Response to comment: In this work, it is claimed that a novel fluorescein probe was prepared but the synthesis was carried out based on a published report.

Response:  We are very sorry for our unclearly describe in the text. Although the synthesis methods which carried out based on a published report, it is totally a new compound of A5 for the first time synthesized in this article. So, we claimed that a novel fluorescein probe was prepared is OK.

  • Response to comment: There are many probes available in the literature for copper sensing and this study lacks the novelty.

Response: More recently, the issue of cuproptosis has received considerable critical attention. The development of faster, convenient and economical method to analyze the copper content in different samples is significantly important for the study of health and environment. There are many probes available in the literature for copper sensing, however as descried in the paper, small molecular fluorescent probes with excellent bio-compatibility and selectivity can also be applied in bio-Imaging is rare. So, it’s high novelty and challenge for us to develop new fluorescent materials for copper sensing.

  • Response to comment: The probe achieved high LOD compared with the published reports.

Response: As described in the text that “And the limit detection of A5 to Cu2+ is much lower than the level which reported in the literature[42]”, the LOD of A5 0.11 µM is lower compared with the literature reported LOD 1.20 µM.

  • Response to comment: Providing the evidence for the formation of Cu-O is important but none of the spectral data highlight this data.

Response: Considering the Reviewer’s suggestion, we have repeated the experiment and found that IR data of the molecular bond of C=O (1705.9) disappeared after the reaction of enough Cu2+(showed in the Figure 4), which can provide the evidence for the formation of Cu-O.

  • Response to comment: Clear explanation of the results and discussion are not presented in the manuscript. The provided spectral and data figures are of low resolutions and are not clearly visible.

Response: We have made correction according to the Reviewer’s comments. We have re-written the results and discussion more in details according to the Reviewer’s suggestion. Also, we provided more quality figures in the revised manuscript.

  • Response to comment: The imaging applications utilized high concentrations of the probe.

Response: Thank you for the suggestion, since we want use test strips experiment to detected Cu2+ qualitatively in aqueous solution, we use the A5 (200.0 μM) and metal ion solutions (200.0 μM) to avoid background interference caused by the fluorescence of filter paper which showed good effect in the revised manuscript.

  • Response to comment: The manuscript has grammatical and formatting errors. Overall, this manuscript is not suitable for the publication.

Response: Thank you for the suggestion, we have checked the grammatical and formatting errors in the text and revised the manuscript to meet the standard of the journal of “biosensors”. Since the A5 molecule in this article as a novel probe was synthesized for the first time by our research group and has good properties in the bioimaging application, we think that this manuscript has good novelty and also suitable for the journal of “biosensors”.

Thank you very much for your comments and suggestions.

We tried our best to improve the manuscript and made some changes in the manuscript. These changes will not influence the content and framework of the paper. And here we did not list the changes but marked up using the “Track Changes” in revised paper.

We appreciate for your warm work earnestly, and hope that the correction will meet with approval.

Once again, thank you very much for your comments and suggestions.

Yours sincerely,

Xin Leng

Round 2

Reviewer 1 Report

The authors have revised the manuscript by following reviewer's comments. The aim and the findings in this manuscript are very interesting. I think that the revised manuscript is now satisfactory and recommended for publication.

Reviewer 2 Report

The revised manuscript entitled “Novel Fluorescence probe toward Cu2+ based on fluorescein derivatives and its bioimaging in cell” was submitted by Xin et al. It is found that the authors took serious effort to revise the reviewer’s comments. Appropriate answer is provided for each raised question. In light of the comments received, I recommend to accept this revised manuscript for the publication in Biosensor Journal.